



**1**  **Investigation of severe dust storms over the Pan-Eurasian**
**2**  **area using multi-satellite observations and ground-based**
**3**  **measurements**

Lu She[1,4], Yong Xue[2,3], Jie Guang[3], Yaihui Che[1,4], Cheng Fan[1,4], Ying Li[1,4], Yanqing
Xie[1,4]
[1]State Key Laboratory of Remote Sensing Science, jointly sponsored by the Institute of Remote Sensing
and Digital Earth of Chinese Academy of Sciences and Beijing Normal University, Institute of Remote
Sensing and Digital Earth, Chinese Academy of Sciences, Beijing 100101, China
[2]Department of Electronics, Computing and Mathematics, College of Engineering and Technology,
University of Derby, Derby DE22 1GB, UK
[3]Key Laboratory of Digital Earth Science, Institute of Remote Sensing and Digital Earth, Chinese
Academy of Sciences, Beijing 100094, China
[4]University of Chinese Academy of Sciences, Beijing 100049, China
*Correspondence to*: Professor Yong Xue (yx9@hotmail.com)
**Abstract.** The deserts in East Asia are one of the most influential mineral dust source regions in the
world. Large amounts of dust particles are emitted and transported to distant regions. A super dust storm
characterized by long-distance transport occurred over the Pan-Eurasian Experiment (PEEX) area in
early May 2017. In this study, multi-satellite/sensor observations and ground-based measurements
combined with the HYbrid Single Particle Lagrangian Integrated Trajectory (HYSPLIT) model were
used to analyse the dynamical processes of the origin and transport of the strong dust storm. The optical
and microphysical properties of the dust particles were analysed using Aerosol Robotic Network
(AERONET) measurements. From the multi-satellite observations, the dust storms were suggested to
have originated from the Gobi Desert on the morning of 3 May 2017, and it transported dust
northeastward to the Bering Sea, eastward to the Korean Peninsula and Japan, and southward to southern
Central China. The air quality in China drastically deteriorated as a result of this heavy dust storm; the
$PM_{10}$ (particulate matter less than 10 mm in aerodynamic diameter) concentrations measured at some air
quality stations located in northern China reached 4000 μg/m³. During the dust event, the maximum AOD
values reached 3, 2.3, 2.8, and 0.65 with sharp drops in the extinction Ångström exponent (EAE) to 0.023,
0.068, 0.03, and 0.097 at AOE_Baotou, Beijing, Xuzhou-CUMT, and Ussuriysk, respectively. The dust
storm introduced great variations in the aerosol property, causing totally different spectral single-
scattering albedo (SSA) and volume size distribution (VSD). The combined observations revealed



comprehensive information about the dynamic transport of dust and the dust affected regions, and the
effect of dust storms on the aerosol properties.
**1.    Introduction**

Dust storms are prevalent in East Asia due to the large scale of deserts. Large amounts of dust

particles are emitted from the deserts in western/northern China and southern Mongolia every year,
especially in the spring (Shao et al., 2011). As one of the major mineral dust sources on Earth, the annual
dust emissions of eastern Asia reach approximately 25% of the total global dust emissions (Ginoux et al.,
2004). These massive emissions produce significant influences on the Earth's radiation balance, climate,
ambient air quality and human health (Goudie, 2009;Shao et al., 2011;Rodríguez et al., 2012). Dust
aerosols exert both direct and indirect effects on the climate system. Dust can directly scatter and absorb
solar radiation over ultraviolet, visible, and infrared wavelengths, resulting in positive or negative forcing
(Rosenfeld et al., 2001;Tegen, 2003). Dust is also involved in cloud formation and precipitation processes
and can alter the albedo of snow and ice surfaces, thereby causing indirect effects on the Earth's energy
budget (Rodríguez et al., 2012;Rosenfeld et al., 2001;Bangert et al., 2012).

Due to the long-distance transport of dust plumes (Zhu et al., 2007), dust particles can alter the

atmospheric conditions in source regions and affect the regional- and global-scale climate (Goudie, 2009).
It has been suggested that the dust from the Taklimakan and Gobi Deserts can travel thousands of miles,
thereby affecting large areas of China (Wang et al., 2013;Lee et al., 2010;Chen et al., 2015;Tan et al.,
2012), South Korean and Japan (Mikami et al., 2006), and even the Northern Pacific Ocean and North
America (Fairlie et al., 2007;Creamean and Prather, 2013;Guo et al., 2017). Dust storms cause poor air
quality and visibility over both origin regions and transport regions and have severe effects on the human
health and environment (Goudie, 2009;Lee et al., 2010). Desert dust is the main contributor to aerosol
loading and PM (particulate matter) mass concentrations in China during the spring season (Wang et al.,
2013). During heavy dust outbreaks, $PM_{10}$ (PM less than 10 mm in aerodynamic diameter) mass
concentrations can even reach 20 exceedances of the internationally recommended limit values in
northern China. Moreover, dust particles can interact with anthropogenic pollution and smoke, causing
air conditions with greater complexity (Dall'Osto et al., 2010).

Many studies have been carried out to study different aspects of dust plumes from deserts using





satellite data, ground-based observations and model simulations (Badarinath et al., 2010;Wang et al., 2013;Teixeira et al., 2016). On the one hand, many studies analyse the chemical composition and source of dust and investigate the radiative effects of dust. These studies focus on the contributions of desert dust to aerosol optical and micro-physical properties to obtain a better understanding (Alam et al., 2014;Basha et al., 2015;Srivastava et al., 2014). On the other hand, some studies are concerned with the long-distance transport of dust plumes using satellite observations and/or model simulations (Huang et al., 2008;Guo et al., 2017;Athanasopoulou et al., 2016) based on the combined use of different data sources to analyse dust formation and transport in depth.

Recently, a heavy dust storm swept through northern China and southern Mongolia from 3 to 8 May 2017. Influenced by the wind, the dust storm spread across southeastern Russia and even reached the Bering Sea on 7-8 May 2017. This dust event exerted a large-scale influence and caused severe air quality problems, especially in northern China. Based on multi-satellite observations and ground-based measurements, the dynamics and the effects of this severe dust storm on the local aerosol properties were deeply investigated. Satellite observations were used to capture the transport of dust. The Ozone Monitoring Instrument (OMI) aerosol index (AI) was used to provide comprehensive information about the absorbing aerosol distribution. Cloud-Aerosol Lidar and Infrared Pathfinder Satellite Observation (CALIPSO) data were used as an ancillary data source to monitor the aerosol type as well as the vertical distribution of the dust particles. The Air Resources Laboratory's HYbrid Single-Particle Lagrangian Integrated Trajectory (HYSPLIT) model was used to generate back trajectories to identify the dust source. Ground-based measurements were collected as a complement to characterize the dust-affected areas and analyse the variations in aerosol properties caused by the dust storm. This study aims to present a large-scale investigation and comprehensive insight into the long-distance transport of the dust event.

## 2. Data and methods

### 2.1 General description of the study area

Fig. 1 shows the topography of the study area. The deserts in China and Mongolia, where an abundance of dust events occur, constitute the second-largest dust source in the world. During the spring, the Gobi region is affected by the Mongolian cyclones, which is the main factor to the severe Asian dust storms (Shao et al., 2011).





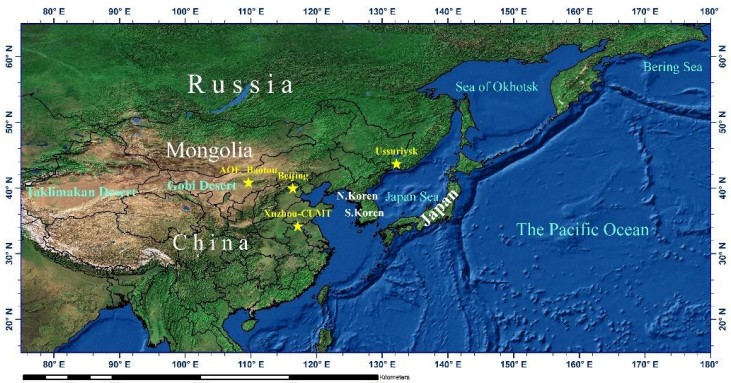


**Fig. 1. Study area of this analysis of dust events. The yellow stars represent four AERONET stations.**
**2.2   Himawari-8 data**

The Himawari-8 (H8) satellite was launched on 7 October 2014 by the Japan Meteorological

Agency (JMA). It started operation on 7 July 2015. The Advanced Himawari Imager (AHI) onboard H8
can provide multi-spectral observations with a high spatial resolution and high frequency. It has 16
channels with a spatial resolution of 0.5-2 km, including 3 visible (VIS) channels (0.47, 0.51, 0.67 μm),
3 near-infrared (NIR) channels, and 10 mid and thermal infrared channels. The AHI level 2 calibrated
data provided by JMA have a spatial coverage of 120° by 120° centred at 0° N, 140° E, and the
observation area includes most of eastern Asia, Australia and the Pacific Ocean. In addition, the AHI
provides full-disk observations every 10 minutes; this provides us with wide-swath, high-frequency
observations to characterize the dust transport. Here, AHI level 2 calibrated data provided by the JMA
and downloaded from the Japan Aerospace Exploration Agency (JAXA) Earth Observation Research
Center (EORC) were used (downloaded from http://www.eorc.jaxa.jp/ptree/terms.html).
**2.3   OMI/Aura**

The OMI sensor aboard the Aura satellite measures the Earth in the ultraviolet (UV) and visible

spectra (270-550 nm) with a wide swath. The observations of the UV spectra make the OMI data suitable
for studying aerosol absorption in the UV spectrum. The OMI provides a parameter called the UV aerosol
index (UV-AI), which is a qualitative parameter that detects UV-absorbing aerosols. The UV-AI is
sensitive to absorbing aerosols, including mineral dust, black carbon, and biomass burning aerosols (Eck
et al., 2001). Therefore, the UV-AI can be used to identify aerosol types through positive values for dust
and biomass burning particles and near-zero or positive values for clouds and weakly absorbing aerosols



(Torres et al., 2007). In addition, the UV-AI can be obtained under both cloudy and cloudless conditions;
the surface reflectance also has no impact on the UV-AI, which can detect absorption by aerosols over
highly reflective surfaces (Torres et al., 2007). Since this dust event occurred in May, a high UV-AI can
be a good indicator of high dust aerosol loading when combined with CALIPSO observations, as Aura
and CALIPSO have similar equatorial crossing times. Here, level 3 OMI UV-AI data were used, which
have a spatial resolution of 0.125°*0.125°.

### 2.4    CALIOP/CALIPSO

The Cloud-Aerosol Lidar with Orthogonal Polarization (CALIOP) instrument on board the
CALIPSO satellite provides vertical profiles of the elastic backscatter at two wavelengths (532 nm and
1064 nm) during both day and night. The CALIOP payload also provides linear depolarization at 532 nm
that can used to identify dust aerosols since dust aerosols have a high linear depolarization ratio due to
their non-sphericity. Aerosol types are also provided in the CALIPSO aerosol product. The CALIPSO
algorithm defines six aerosol types, including smoke, dust, polluted dust, clean continental, polluted
continental, and marine (Omar et al., 2009;Omar et al., 2013). It has been suggested that the CALIPSO
aerosol classification works well in most cases (Wu et al., 2014). It should also be considered that the
accuracy of aerosol detection is decreased over highly reflected land surfaces such as deserts and snow-
covered regions, and there is no aerosol information from passive sensors (e.g., OMI) during the night-
time. Here, CALIPSO level 2 vertical feature mask (VFM) aerosol layer products were used to provide
independent information about dust aerosols, especially for the night-time, as the signal-to-noise ratio
during the night-time is better than that during the daytime for CALIPSO (Liu et al., 2009). The VFM
products have a vertical resolution of 30 m below 8.2 km, 60 m for 8.2-20.2 km, and 180 m for 20.2-
30.1 km (Winker et al., 2007). The dust information, especially regarding the vertical distribution and
dust layer height, were analysed using CALIPSO VFM data.

### 2.5    AERONET data

The Aerosol Robotic Network (AERONET) is a ground-based remote sensing aerosol network
(Holben et al., 1998) that provides spectral AOD and inversion products derived from direct and diffuse
radiation measurements by Cimel sun/sky-radiometers (Dubovik et al., 2006). The inversion products
includes both microphysical parameters (e.g., the size distribution and complex refractive index) and



radiative properties (e.g., the single-scattering albedo and phase function) (Dubovik et al., 2006).
In this study, Level 1.5 cloud screened data including both sun direct data (Version 2 and Version 3) and
Inversion data (Version 2) from four AERONET sites in the study area were used to analyse the temporal
variations in aerosol properties, including the AOD, the extinction Ångström exponent (EAE), volume
size distribution (VSD), and single-scattering albedo (SSA). Fig. 1 shows the locations of the four sites
(yellow stars), namely, AOE_Baotou, Beijing, Xuzhou-CUMT, and Ussuriysk.

**2.6  PM measurements**

There are thousands of air quality stations over China that can provide hourly PM measurements
during both the daytime and the night-time. In addition, the measurements are free from the influences
of clouds, making it a perfect complement to AERONET observations and satellite observations, as few
AERONET stations provided useful observations over China during May 2017. Ground-based
measurements of the PM mass concentration over the Chinese mainland were collected to illustrate the
dust-affected areas and further analyse the transport of the dust plume. Furthermore, the temporal
variations in the PM concentrations at 14 typical stations were analysed in detail to examine the
propagation of dust particles in different directions. Detailed information about these 14 air quality
stations is given in Table 1.

**Table 1. The cities and locations of the 14 air quality stations**

| City (Site) | Longitude | Latitude | City (Site) | Longitude | Latitude |
|---|---|---|---|---|---|
| Bayannao'er (BYN) | 107.5936 | 40.916 | Shanghai (SHS) | 121.536 | 31.2659 |
| Changsha (CSS) | 112.9958 | 28.3586 | Taiyuan (TYS) | 112.5583 | 37.7394 |
| Chengde (CDS) | 117.9664 | 40.9161 | Tianshui (TSS) | 105.7281 | 34.5814 |
| Guangyuan (GYS) | 105.8153 | 32.4246 | Tongliao (TLS) | 122.2603 | 43.6267 |
| Heihe (HHS) | 127.4961 | 50.2486 | Weihai (WHS) | 122.0508 | 37.5325 |
| Huhhot (HHT) | 111.7277 | 40.8062 | Zhengzhou (ZZS) | 113.6113 | 34.9162 |
| Jiangchang (JCS) | 102.1878 | 38.5247 | Zhongwei (ZWS) | 105.18 | 37.0172 |




### 2.7 HYSPLIT

The NOAA HYSPLIT model developed by NOAA's Air Resources Laboratory was employed (Draxler and Rolph, 2013). It is widely used for computing air mass forward/backward trajectories to analyse the transport of air/pollution parcels. The start/end points as well as the time of the HYSPLIT computation can be set depending on your interest. Here, HYSPLIT was used to generate air mass backward trajectories to trace the air movement.

### 3. Results

#### 3.1 Origin and transport of the dust event

Fig. 2 shows the spatial distribution of the UV-AI over East Asia from 3 to 8 May 2017 obtained from the OMI-Aura observations. High AI values can be observed over northern China, especially over Inner Mongolia on 3 May, northeastern China on 4-5 May, and southeastern Russia on 5-6 May. The maximum AI values even exceed 3, indicating the existence of a large area with a high loading of absorbing aerosols. From multi-day images, temporal variations in the AI distribution were clearly observed, and the regions with high AI values were moved towards the east and northeast. The dust storm initially developed over western Inner Mongolia (~ 40° N, 100° E) on 3 May 2017 (see Fig. 2a) and then swept through the North China plain on 4 May 2017 due to a strong easterly wind, and the dust storm reached northeastern China (~50° N, 125°E) within one day. On 5 May, the dust plume was transported to the western Sea of Okhotsk (~56°N, 140°E). For the next two days, the elevated dust plume travelled across the Sea of Okhotsk and finally reached the Bering Sea on 7-8 May (see Fig. 2e-f). The OMI-AI effectively revealed the long-distance transport of the strong absorbing aerosols that originated from the Gobi Desert.

To be sure that the high AI values were caused by dust aerosols, CALIPSO observations that passed through the dusty regions during the night-time were employed to provide aerosol type and vertical distribution information of the dust plume. Fig. 3 shows the overpass trajectory of the CALIPSO observations employed in this study during 3-8 May. Fig. 4 depicts the vertical distributions of the aerosol types and their corresponding overpass trajectories. As Fig. 4b illustrated, large numbers of elevated dust aerosols were distributed over Inner Mongolia and Shanxi Province (from ~40°N -~32° N) on 4 May. As the dust plume travelled eastward and northeastward, a dominant, thick dust layer was observed over





the southeastern Russia, northeastern China and Yellow Sea regions on 5 May (Fig. 4c). Especially over
southeastern Russia, the dust layer was thick and distributed from the surface to a height of 10 km. In the
following several days, the elevated dust particles were transported northeasterly and proceeded to the
Sea of Okhotsk (Fig. 4d) and Russia's remote Kamchatka Peninsula (Fig. 4e) before finally reaching the
Bering Sea (Fig. 4f).
Moreover, a part of the aerosol layer was marked as a polluted dust subtype by the VFM product
over Central China on 4 May and over the region of northern China on 6 May. This may be explained by
the mixture of dust and anthropogenic pollution during the movement of the dust plume. In addition, dust
marine aerosol layers over the ocean were also detected on 6-8 May (Fig. 4d-f).

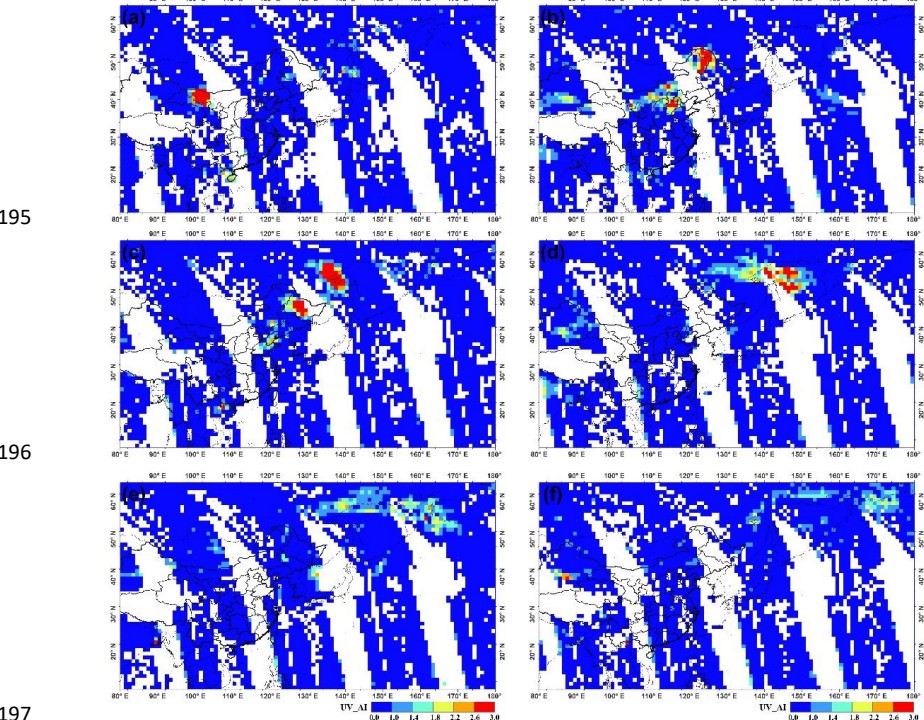




**Fig. 2(a-f). Spatial distributions of the OMI UV aerosol index from 3 to 8 May 2017.**

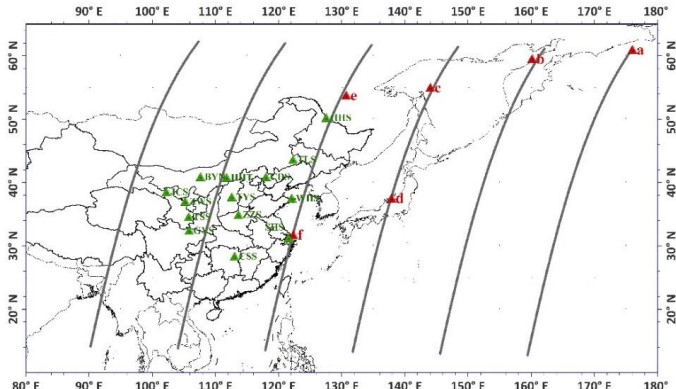


**Fig. 3. The overpass trajectories of CALIPSO observations (grey lines) during 3-8 May, the locations of the**
**air quality stations (green triangles), and the end points of the HYSPLIT computation (red triangles).**

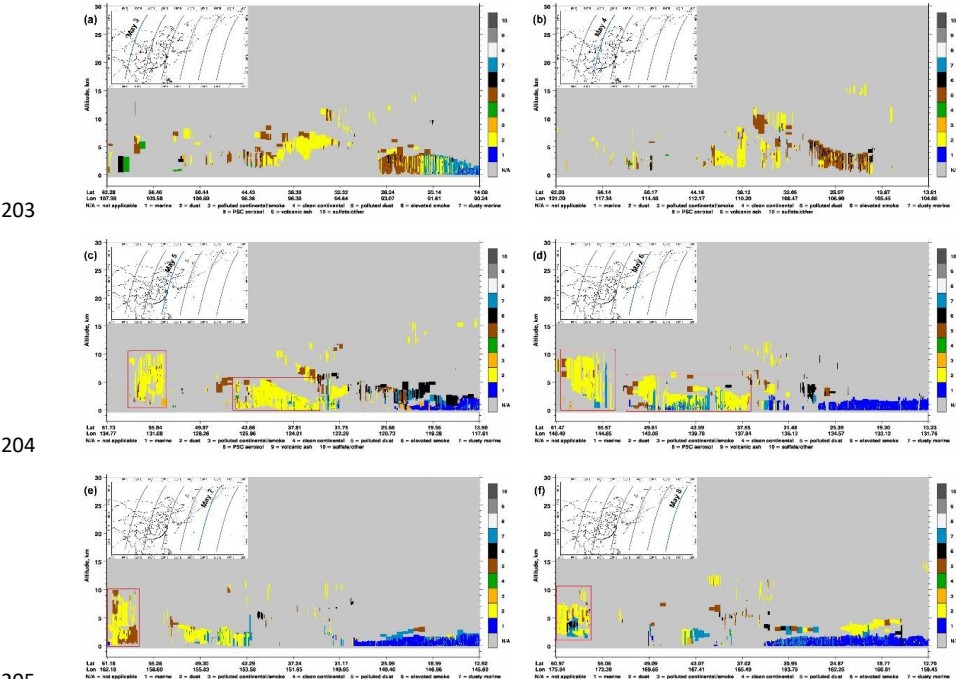




**Fig. 4. a-f. CALIPSO aerosol subtypes on 3-8 May 2017. The grey and blue lines in the map represent the**
**orbit tracks used in this work, while the blue line is the corresponding overpass trajectory of the aerosol**
**subtype.**

Fig. 5 shows the backward trajectories at different sources (the red triangles in Fig. 3) within the

dusty regions (the red square in Fig. 4c-f) during 5-8 May 2017. The trajectories are computed at three




different altitudes (1000 m, 2000 m, and 3000 m). The HYSPLIT backward trajectory analysis revealed
that the air masses that reached the Bering Sea (Fig. 5a), the Kamchatka Peninsula (Fig. 5b), and the Sea
of Okhotsk (Fig. 5c) in addition to southeastern Russia (Fig. 5e) and eastern China (Fig. 5f) were derived
from the Gobi Desert. The This result is consistent with that from the OMI-AI and CALIPSO aerosol
type information, providing clearer insight into the sources as well as the movements of the dust particles.

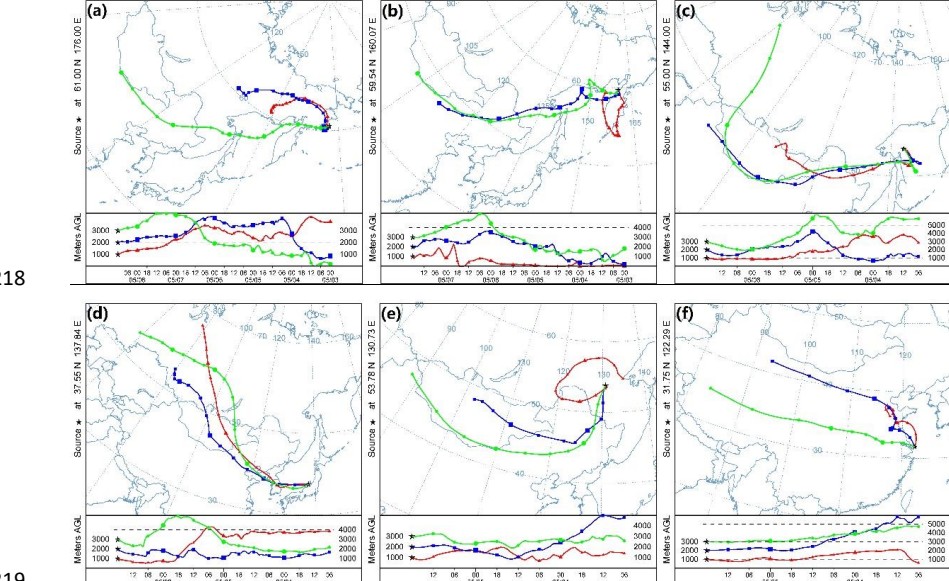



**Fig. 5. Backward trajectories derived from the HYSPLIT model at different locations during 5-8 May 2017:**
**(a) 132-h back trajectories ending at the Bering Sea on 8 May, (b) 108-h back trajectories ending at the**
**Kamchatka Peninsula on 7 May, (c-d) 84-h back trajectories ending at the Sea of Okhotsk and the Sea of**
**Japan on 6 May, respectively, and (e-f) 60-h back trajectories ending at southeastern Russia and the Yangtze**
**River estuary region on 5 May, respectively.**

As dust plumes usually move fast with a high temporal variation, polar-orbiting satellites can
typically provide only one or two observations per day. Therefore, it is potentially impossible to detect
the rapid movements of dust events using polar-orbiting observations, as some dust activity would be
missed due to the limited pass time and dust deposition. Geostationary satellites can provide high-
frequency observations over large areas and have unique advantages for obtaining the comprehensive
spatial-temporal variations of dust events. For a better view of the transport of the dust plume, the high-
temporal-resolution observations from the Himawari-8/AHI were used. A time series of true-colour
composite images on 3 May and 4 May were analysed for more detailed information about the dust



evolution. Fig. 6 shows the composite images over a 3-h interval from 03:00 to 09:00 UTC on 3 May.
The results suggest that the strong dust storm originated from the western part of the Gobi Desert and
was formed by several distinct dust clusters. In the morning on 3 May, only a small area was covered by
a dust plume in the Gobi Desert as the dust storm was continuously increasing and quickly moving. On
the one hand, the dust plume over southwestern Inner Mongolia moved along the edge of the Qinghai-
Tibet Plateau and then finally reached the northern Sichuan basin (Fig. 6c). On the other hand, massive
dust storms travelled along the China-Mongolia border with a continually increasing dust intensity and
quickly moved towards the northeast and east. The dust plume moved northeastward reached the border
of China, Mongolia and Russian on the afternoon of 3 May. As the dust plume moved eastward, it arrived
in the North China Plain and northeastern China on the morning of 4 May (Fig. 7), causing more than 10
provinces in northern China to be covered by a dust plume. In addition, in the late afternoon of 4 May
2017, another dust storm was found that originated from northern Inner Mongolia (Fig. 7e-f) that was
quickly transported eastward due to strong westerly winds. High-frequency observations from the AHI
presented more information about this dust event, revealing a continuous dust storm and several different
transport directions, including southeastward, eastward and northeastward. The longest-distance
transport occurred in the northeastward direction, as OMI-AI and CALIPSO-VFM illustrated in the
previous section, and the dust finally arrived at the Bering Sea.

**3.2  PM characterization in China during the dust event**
In this section, the temporal variations in the $PM_{2.5}$ and $PM_{10}$ mass concentrations over mainland
China were deeply analysed. The dust plume often caused a high aerosol loading and high PM
concentration, especially $PM_{10}$. Fig. 8 depicts the $PM_{10}$ concentration distribution over mainland China
over a 12-h interval from 06:00 a.m. on 5 May to 06:00 p.m. on 7 May (Beijing time). Interestingly,
southeastward transport was revealed through the intensive PM concentration measurements, while it
was almost missed by most of the satellite observations because central and eastern China were covered
by a huge cloud during 5-7 May. The high $PM_{10}$ concentration was mostly distributed over 35-40° N at
06:00 on 5 May (Fig. 8a); meanwhile, after 12 hours, the dust plume moved to Shandong Peninsula and
Henan Province and further affected Central China on 6 May; two days later, the dust events were found
in most stations of eastern and central China (Fig. 8 c-f). The southward propagation of this dust event




caused a high PM$_{10}$ concentration (>500) in south-central China (e.g., Hunan Province) as well as the
eastern coastal areas including the Shandong Peninsula, Jiangsu Province and the Yangtze River Delta.

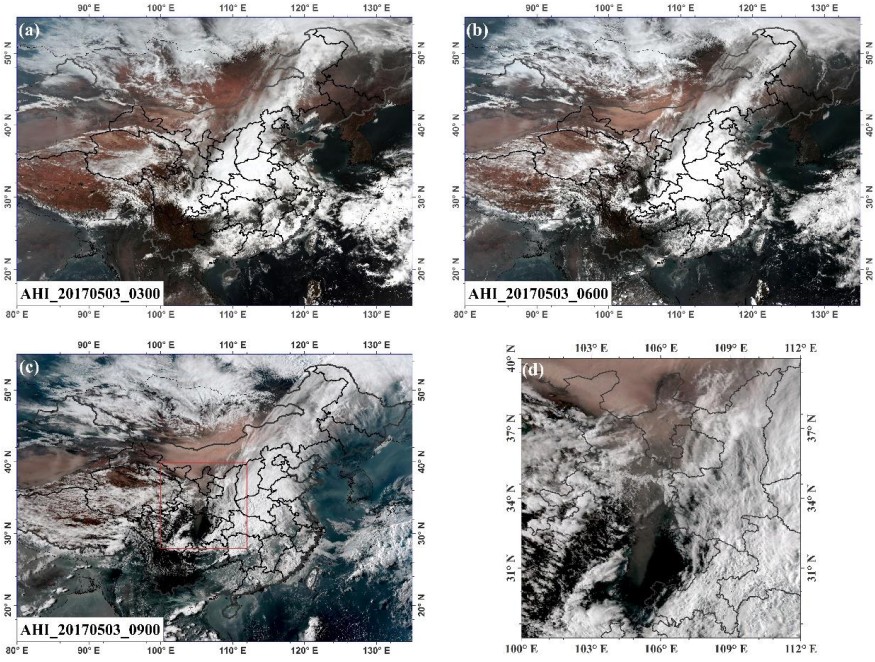

**Fig. 6. True-colour composite images of mainland China (a-c) from AHI data over a 3-h interval on 3 May**
**2017. (a) 03:00, (b) 06:00, (c) 09:00, and (d) the area of the red square frame in (c).**

To obtain better insight into the dust evolution, measurements from 14 typical air quality stations
(the green triangles in Fig. 3) situated within the origin and transport areas of the dust were analysed in
detail. As the PM concentration was measured during both the daytime and the night-time, the data can
provide much more information about this continuous dust plume. Fig. 9 shows the PM temporal
variations along three different dust transport directions during 2 to 7 May, including the northeastward
propagation (a), southward propagation (b) and southeastward propagation (c). It is clearly observed that
both the PM$_{2.5}$ and the PM$_{10}$ were increasing dramatically, and the PM$_{10}$ showed much larger increments
than the PM$_{2.5}$ during this dust event from all three figures.
As Fig. 9a illustrates, the sharp increase in the PM mass concentration was first observed at station
BYN on the morning on 3 May, followed by the stations at CDS (23:00 UTC on 3 May) and TLS (8:00
UTC on 4 May), and reached the northeastern-most city, namely, Heihe (HHS) (06:00 UTC on 4 May).
The maximum value of PM$_{10}$ concentration at BYN reached 4333 μg/m$^3$ on 4 May.   And continuing
sharp increase in the PM$_{10}$ concentration were observed at those sites, indicating continuous outbreak of
dust storms. Cities in northeastern China were deeply affected by the transported heavy dust storms, high
PM$_{10}$ concentrations occurred successively at those sites. These drastic changes in the PM$_{10}$ are in
agreement with the dust movements revealed from the satellite observations.

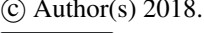



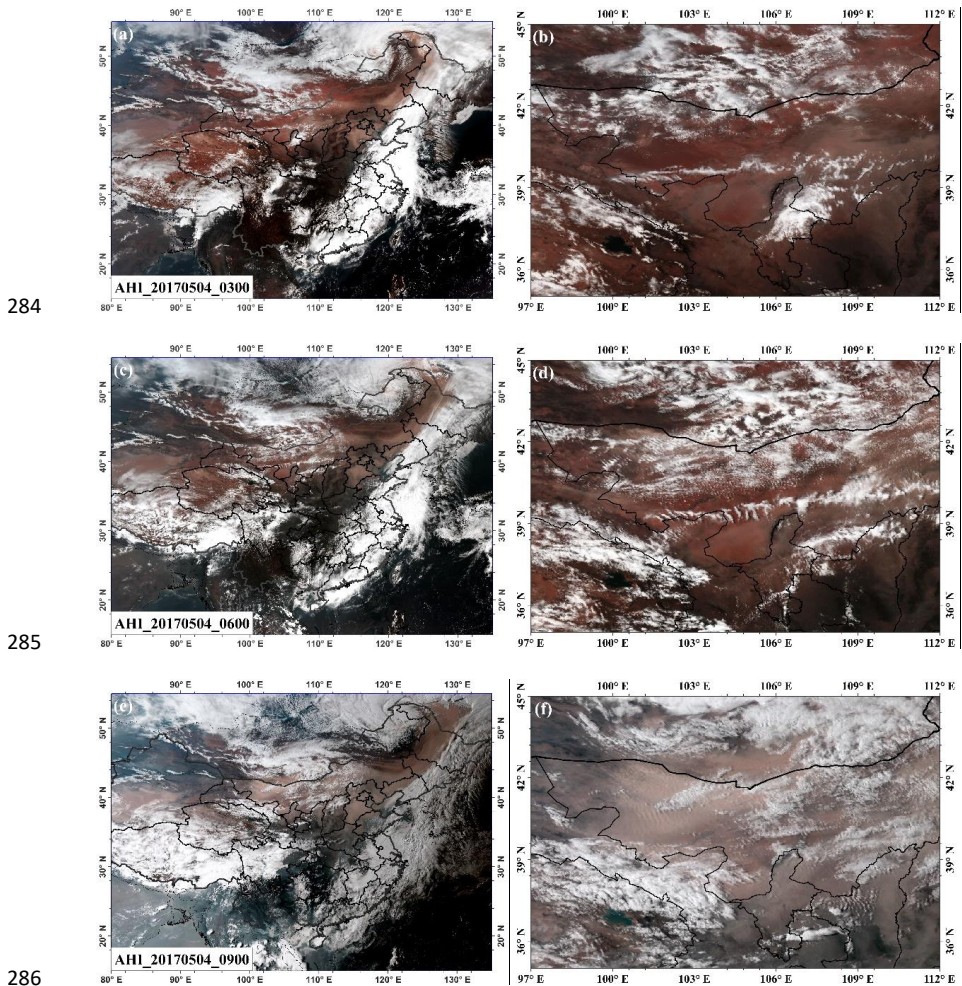

**Fig. 7. True-colour composite images of mainland China (a, c, and e) and western Inner Mongolia and the surrounding areas (b, d, and f) from AHI data over a 3-h interval from 03:00 UTC to 09:00 UTC on 4 May 2017. (a) and (b) are at 03:00, (c) and (d) are at 06:00, and (e) and (f) are at 09:00.**

PM measurements at 4 stations distributed along the eastern edge of the Qinghai-Tibet Plateau, including JCS, ZWS, TSS and GYS, are shown in Fig. 9b. Within one day, the dust plume was transported across Gansu and reached GYS, which is located in the Sichuan Basin. This transport was also revealed by the high-frequency AHI observations (Fig. 6c and d), although it is not as noticeable.



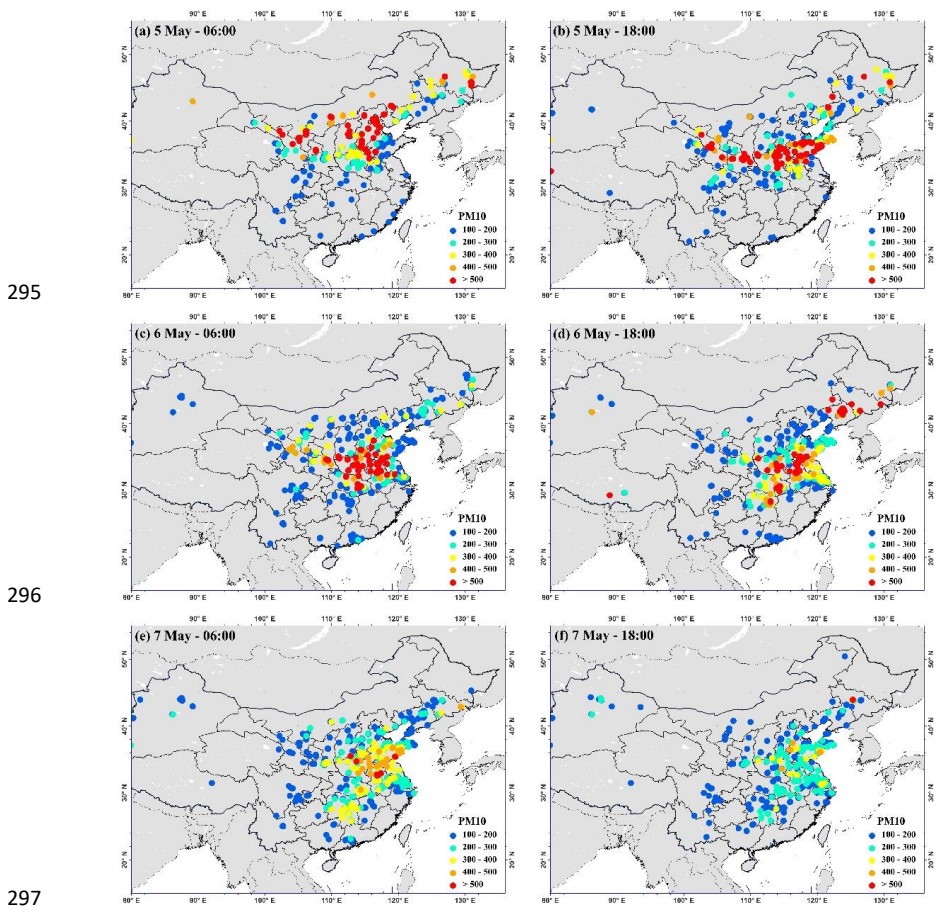

**Fig. 8. PM10 mass concentrations measured by ground-based air quality stations in mainland China over a 12-h interval from 06:00 on May 5 to 18:00 on May 7.**

Fig. 9c displays the PM concentration variations over the cities located in Central China, including Taiyuan (TYS), Zhengzhou (ZZS), and Changsha (CSS). The sharp increase in the $PM_{10}$ concentration with a very slight rise in the $PM_{2.5}$ concentration indicates that the dust plume travelled to southern Central China and caused a bad air quality there. In addition, high PM concentration were observed in the coastal areas of eastern China, as Fig. 10 shows. Note that the increases of PM10 are much larger than the increments of PM2.5 in those stations, suggesting that the dust particles were transported to southern and eastern China.

To confirm this southward propagation of dust, the backward trajectories ending at GYS, CSS, and SHS were analysed by HYSPLIT, as shown in Fig. 11. The trajectories are computed at three different


altitudes (500 m, 1000 m, and 1500 m). As the trajectories illustrate, the northwestern air masses at all
three locations originated from sources in the Gobi Desert. Thus, dust could be the main reason for the
sudden jump in the PM concentrations. The back trajectories at the three sites computed from HYSPLIT
are consistent with our analysis based on PM measurements.

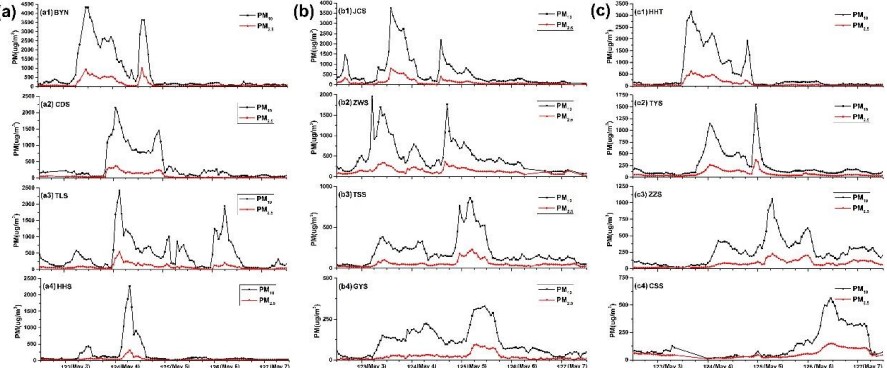

**Fig. 9. Time series of the PM2.5 (red curves) and PM10 concentrations (black curves) during 2-7 May at 14**
**stations in three directions: (a) northeastward propagation, including (a1) BYN, (a2) CDS, (a3) TLS and (a4)**
**HHS, (b) southward propagation, including (b1) JCS, (b2) BYS, (b3) TSS and (b4) GYS, and (c)**
**southeastward propagation, including (c1) HHT, (c2) TYS, (c3) ZZS and (c4) CSS.**
**Fig. 10. Time series of the PM2.5 (red curves) and PM10 concentrations (black curves) during 2-7 May at (a)**
**WHS and (b) SHS.**
**Fig. 11. Backward trajectories derived from the HYSPLIT model at different altitude levels (500 m, 1000 m,**
**and 1500 m) at (a) GYS on 3 May, (b) SHS on 6 May, and (c) CSS on 6 May 2017.**





### 3.3 Aerosol property variations during the dust event

In order to understand the effects of dust storm on aerosol properties, the changes in the aerosol properties at four typical AERONET stations located in the study area were investigated. These four sites are located in different environments; the longitudes of AOE_Baotou, Beijing, and Ussuriysk increase from west to east, and the latitudes of AOE_Baotou, Beijing, and Xuzhou-CUMT decrease from north to south. This can help to illustrate the temporal variations in the aerosol characteristics due to the movement of the dust plume. Several key parameters, including the AOD, EAE, SSA, and aerosol volume size distribution (VSD), were analysed in detail.

The temporal variations in the daily AOD and EAE at the four AERONET sites during dusty and non-dusty days are plotted in Fig. 12 a-d. The maximum AODs at 440 nm caused by the dust storm were 2.96, 2.13, 2.87, and 0.65 at AOE_Baotou, Beijing, Xuzhou-CUMT, and Ussuriysk, respectively. The maximum AOD at Baotou (the westernmost station) was recorded on 2 May 2017 and became lower afterwards with a low EAE value of 0.15. Another increase in the AOD as well as a drop in the EAE occurred on 4 May, and the dust continued for several days. Then, the dust storm moved eastward, and the highest AOD value of 2.13 was observed over Beijing on 4 May 2017. As the dust storm travelled northeastward, Ussuriysk, located in southern Russia, was affected with a slight increase in the AOD (from ~0.25 to ~0.65) and a sharp decrease in the EAE (from ~1 to ~0.1) on 5 May 2017. Xuzhou-CUMT, which is located in southern Central China, was also severely affected by the strong dust on 4-5 May. The maximum AODs occurred at different times at the four sites due to the movement of the dust storm. In addition, there are obvious negative correlations between the AOD and EAE during the dust event. The dust storm brought numerous large particles, causing the low EAE and high extinction properties.

As one of the most important properties affecting aerosol radiative forcing, aerosol absorption also exhibits huge variations. The SSA is strongly related to absorption/scattering characteristics. Fig. 13 shows the variability of the spectral SSA before, during and after this dust event, and it is compared with the monthly average. The SSA at longer wavelengths (e.g., 675, 870, and 1020 nm) at AOE_Baotou varied from ~0.8 to ~0.9 during non-dusty days (1 May and 6 May), and the monthly average of $SSA_{675nm}$ was approximately 0.9, while $SSA_{675nm}$ increased to 0.97-0.98 during dust days (2 and 4 May). In addition, the spectral behaviour of the SSA showed significant differences. The SSA increased with the wavelength on 2 May and 4 May. Especially on 4 May, the SSA largely increased from 440 nm to 675 nm (from 0.93



to 0.98), and the dSSA (dSSA=$SSA_{870nm}$-$SSA_{440nm}$) also increased to 0.07. According to Dubovik et al.
(2002), mineral dust aerosols tend toward a dSSA value greater than 0.05. In contrast, the monthly
average of the spectral SSA as well as the spectral SSA during non-dusty days obviously decreased with
an increase in the wavelength. The high SSA and increasing spectral behaviour indicates that aerosol
particles are dominated by large particles with strong scattering. However, it was noticed that the
$SSA_{440nm}$ on 2 May was high with low absorption. This could be explained by the mixture of dust aerosols
with large amounts of anthropogenic aerosols from industrial emissions, which are more absorbent.

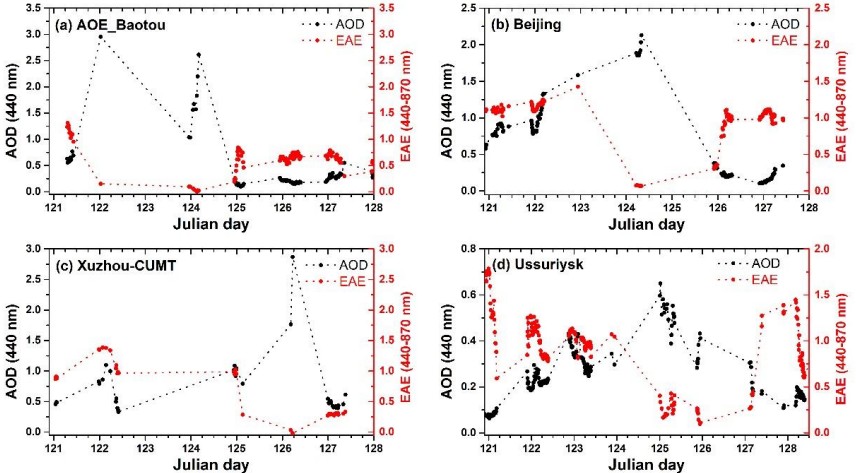

**Fig. 12. Variations in the AOD (440 nm) and Ångström exponent (440-870 nm) at (a) AOE_Baotou, (b) Beijing,**
**(c) Xuzhou-CUMT, and (d) Ussuriysk during 1-8 May 2017.**
Similar properties can be observed over Beijing, as the dust over both Baotou and Beijing have
similar sources. However, there are still a few differences. The monthly average of the spectral SSA in
Beijing was lower than that in AOE_Baotou, and an opposite spectral dependence was observed between
these two sites. Baotou was affected by a greater quantity of industry emissions than Beijing, as it is a
heavy industry city. In addition, it also suffered from additional dust due to its geographical location.
The VSD variation showed a more obvious distinction between dusty and non-dusty days. As Fig.
13 illustrates, the particle volumes of fine-mode aerosols are comparable with those of coarse-mode
aerosols in Beijing and Baotou during non-dusty days. The strong dust storm caused a dramatic increase
in coarse-mode particles compared with non-dusty days. The volume median radius also showed
differences between dusty and non-dusty days; the VSD peaks increased with the AOD due to the dust





storms, and the peak occurred at radii of ~2 μm with peak values of 1.05 and 1.8 on 4 May at Baotou and
Beijing, respectively. Meanwhile, no significant variation was observed for fine-mode particles. It is
observed that the volumes of both fine- and coarse-mode particles were large at AOE_Baotou on 2 May
due to the combination of fine-mode aerosols with dust particles. This also explains the spectral SSA
behaviour on that day.

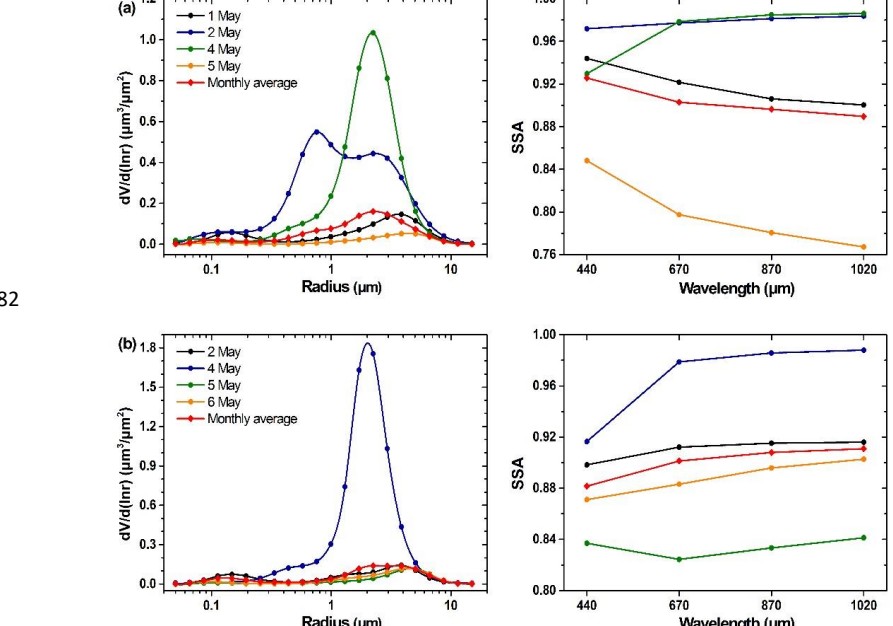



**Fig. 13. Variations in the daily aerosol volume size distribution and spectral SSA during the dust event at (a)**
**AOE_Baotou and (b) Beijing. Different colours represent different days, and the red curves represent the**
**average VSD and SSA in May 2017.**
**4.    Conclusions**

In this study, we described a strong dust storm that occurred in northern China and Mongolia in

early May 2017. The origin and transport were investigated using multi-satellite data (including OMI,
CALIPSO, and AHI), ground-based measurements (including PM measurements and AERONET
observations), and HYSPLIT model computations. Benefiting from the high frequency of geostationary
satellite observations, the rapid spatial-temporal variations in the dust plume were captured, including
the continuous dust storms originating from the Gobi Desert region and different transport directions



over China region. The OMI-AI and CALIPSO observations during the night-time provided more
comprehensive information with larger coverage for the large-scale transport and vertical distribution of
the dust plume. Intensive measurements (in both time and space) of the PM concentration revealed
additional details when the region was covered by thick clouds and CALIPSO covered limited
observation areas. The backward trajectories computed from the HYSPLIT model also confirmed the
directions of dust transport. From the combined observations, this severe dust storm was suggested to
have originated from the Gobi Desert, due to the strong winds, the continuous dust storms travelled to
three different directions and affected large areas of China, including northern China, southeast China,
and even Central China. In addition, southern and eastern Russia and the Bering Sea were influenced by
the long-distance transport of the strong dust plume. The aerosol properties (AE, SSA, and VSD) have
changed greatly during the dusty days, numerous large particles contributed to strong scattering and
extinction. Overall, the combined observations of satellite- and ground-based data contributed to the
comprehensive monitoring of the origin and long-distance transport of the dust storms, providing
complete information on the spatial-temporal distribution.
**Acknowledgements**

This work was supported in part by the National Natural Science Foundation of China under Grant

no. 41471306. We gratefully acknowledge the support by the Strategic Priority Research Program of the
Chinese Academy of Sciences (Grant No. XDA19070202) and the Open Fund of the State Key
Laboratory of Remote Sensing Science (Grant No. OFSLRSS201703). The OMI and CALIOP data were
obtained from NASA. The AHI data were supplied by the P-Tree System, Japan Aerospace Exploration
Agency (JAXA) (http:/ Himawari-8 data /www.eorc.jaxa.jp/ptree/terms.html). The PM data used in this
work were acquired from the China Meteorological Administration. Many thanks are due to the principal
investigators of the AERONET sites used in this paper for maintaining their sites and making their data
publicly available. We would also like to thank the anonymous reviewers for their valuable comments,
which greatly improved the quality of this manuscript.

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
