# Peer review of "Investigation of severe dust storms over the Pan-Eurasian area using multi-satellite observations and ground-based measurements"

_Natural Hazards and Earth System Sciences, 2018_

## Referee Comment (RC1) · Anonymous Referee #1 · 20 Jun 2018

This manuscript describes a severe dust episode originating from Gobi desert on early May 2017. The authors present the event properties based on satellite, in-situ and model backtrajectory data. The manuscript is well written and the data are clearly described. However I do not recommend publication in NHESS. The reason is that at this stage it looks more like a report rather than a scientific paper and there is no clear justification of the contribution of this study to the relevant literature (e.g. unique properties of the particular event, explanation of the system behaviour, impact, etc). A simple presentation of measurements does not really contribute to our understanding on these events nor to the improvement of forecasting or mitigation activities. Similar measurements and observations are routinely performed worldwide. For example the

origin and the evolution of this specific event has been forecasted by operational atmospheric dust models (e.g. http://www.bsc.es/ess/bsc-dust-daily-forecast ) so there is really no need to perform HYSPLIT backtrajectories.
* * *

---

## Referee Comment (RC2) · Anonymous Referee #2 · 20 Jun 2018

In this paper, authors use multi-satellite observations and ground-based measurements to analyses a strong dust storm occurred in East Asia during 3 - 8 May 2017, the long-distant transport of the strong dust storms and the properties of dust aerosols were analyzed. The paper investigated the sources and different transport directions of the dust storms from different satellite observation (OMI, CALIPSO, and AHI) and particle matter (PM) measurements from ground-stations, and the HYSPLIT model were used to calculate the backward trajectories of air masses. The aerosol properties and its variation during no-dusty and dusty days were compared using AERONET data. The paper is clearly structured and logical. The authors combine advantages of satellite data and ground-based data, giving readers a comprehensive and detailed view

for this dust event, including its transport trajectory, horizontal and vertical properties of storm, and its influence on aerosol properties. It can be expected that the study provides a useful contribution to dust transport and related to this Journal. However, the language of the paper requires some improvements. There are some sentences that are unclear or too long to follow. There are also some redundancies that should be removed. But I realize the authors' first language is not English, and this is not a criticism of them. I would recommend publication if my following comments/suggestions can be adequately addressed.

Some comments and questions are given as follows:

Major comments:

1. The core of this paper, in my opinion, is to clearly describe the dust transport process and the dust affected areas. The authors used long length to explain the transport of dust storm based on multi observations, but it would be better to see a more compact analysis with clearer connections between different observations.

2. The authors should define the scientific aims of this study in more detail than what is done in the last paragraph of the introduction.

3. The authors point out that the dust transported to Korean Peninsula and Japan, but I don't see much analysis supporting these findings, especially for Korean Peninsula. Please check this claim more carefully.

4. The authors have also analyzed the aerosol property variation using four AERONET sites measurements. The variations in the AOD (440 nm) and Ångström exponent at four sites are shown, but why just show the VSD and SSA at Beijing and Baotou, what about Xuzhou-CUMT and Ussuriysk?

5. There are some sentences and points which are confusing and invalid, even misleading readers. I suggest authors polish those important sentences to make your analysis more useful and clear.
6. It is hard to read the figures, because some figures are heavily digitized. So I suggest authors to re-plot them or upload un-compressed manuscript.

Detailed comments:

1.Line 53, 'mm' should be '$\mu$m'

2.Fig.2, suggest to use "brown" or other color scheme to represent the UV_AI within 0-1. In addition, the labels on the color bar almost cannot be read! Please enlarge.

3.Fig.3 the PM sites cannot be read. Please enlarge. We can barely read what is written.

4.Fig.4 the orbit tracks is not clearly depicted, please enlarge or just deleted, as the trajectories have been shown in Fig.3

5.Line 191, 'over the region of northern China on 6 May', it seems that the overpass trajectory of 6 May didn't pass over northern China, see fig.4d. Please check it

6.Line 212-215, sentence structure needs to be revised

7.Line 231, 'true-colour' should be 'true color'

8.Fig.6 and Fig.7 are somewhat blurred, it's hard to tell the 'dust clusters' that described in line 235, as well as the dust transport.

9.Line 233-247: This section is a bit confusing, it should be rephrased to make it clearer.

10.Line 262 'caused a high PM10 concentration (>500) in south-central China (e.g., Hunan Province)' It would be better to specify the fig.- rather 'Hunan province', as it's not shown on the map, it is just a new city name to reader.

11.The authors should clearly conclude the transport process of dust, including different transport directions in 'Result' section.

---

## Referee Comment (RC3) · Anonymous Referee #3 · 5 Jul 2018

General comments

The Study presented in this manuscript analyze in details large-scale heavy dust storm during May 2017 over Asia. Airborne dust originated from Gobi desert dispersed in several dust plumes, which propagated for several days in different directions. The authors used diverse sources of observations to generate the knowledge on origin, timing and spatial coverage of the dust storm, overcoming setbacks of one observational system with other sources of measurements, leaving no room for uncertainties in created hypothesis on this event. Scientific significance, scientific and presentation quality are good. Presented subject is of great significance because of the popularity

of the topic, large impact of dust on climate system, but still not well understood and poorly represented in numerical models. Case study described here may be well used in further numerical models development and verification, since it is hard to correctly capture and describe fully any dust storm. This reviewer recommends this manuscript for publishing, after consideration of the following comments.

Specific comments

1) The title mentions in plural "dust storms", but in the manuscript is analyzed one dust storm that dispersed in several dust plumes. In the text is also mixture in mentioning dust storm as single event and dust storms as plural. To avoid confusion the authors should decide to define this event as one dust storm that has divided in several dust plumes or to define this event as severe airborne dust transport, which consists of several dust storms with the same origin. This reviewer suggests defining described event as severe dust storm that has complex multi-plume propagation. Whatever the authors decide, title and the mentioning in the text of the manuscript should be changed accordingly. In the title should be the date of the event, to outstand that the study covers specific study case.

2) In the manuscript there is no analysis of meteorological parameters to be able to understand the atmospheric conditions that produced this severe large-scale dust storm. It is very important, to fully understand the event, to provide information about synoptic situation. To simplify this request it is enough to add the information on surface wind velocity and direction in the source region at the time of dust emission (or surface wind field), and to provide wind fields at representative height and/of geopotential heights (for example 500mb level) in representative times for later days. This would additionally explain the atmospheric circulation that carried dust particles. Data can be used from reanalysis fields.

3) Add information about source of input data for HYSPLIT model that produced backward trajectories.
4) It would be very useful to add an image that presents hypothesis about dust storm propagation in different directions (or mark with arrows in Fig. 1), which is proved using many observations. It is hard to follow in case the geography of the region is not well known.

Technical corrections

1) line 26: change "10 mm" in "10 $\mu$m"

2) line 55: change "10 mm" in "10 $\mu$m"

3) line 59: change "Many studies have been carried out to study different aspects of dust plumes from deserts using..." in "Many studies have been carried out to study different aspects of airborne dust transport from deserts using..."

4) line 120: change "that can used to..." in "that can be used to..."

5) line 123: change "It has been suggested that..." in "It has been evaluated that..."

6) line 136/137: change "The inversion products includes both microphysical parameters ..." in "The inversion products include both, microphysical parameters ..."

7) line 146: change in "... during both, day-time and night-time."

8) line 149/150: change in "...were collected to evaluate the dust-affected areas and to further analyse the transport of the dust plume."

9) line 172: is it correct to have easterly wind in "... swept through the North China plain on 4 May 2017 due to a strong easterly wind,..."? It is not likely to have east wind, maybe west wind, which means that circulation was eastward?

10) line 215: exclude "The" at the beginning of "The This result is ..."

11) line 254: In the following sentence "Fig. 8 depicts the PM10 concentration distribution..." add an information about PM data values, are they hourly average of what? how many stations are considered?

12) Letters in some Figures are too small, and require large zoom to be readable, especially Fig. 2, Fig. 4 and Fig. 9. If possible, use different rearrangement of plots and landscape mode.

---

## Author Response (AR1)

This manuscript describes a severe dust episode originating from Gobi desert on early May 2017. The authors present the event properties based on satellite, in-situ and model back trajectory data. The manuscript is well written and the data are clearly described. However I do not recommend publication in NHESS. The reason is that at this stage it looks more like a report rather than a scientific paper and there is no clear justification of the contribution of this study to the relevant literature (e.g. unique properties of the particular event, explanation of the system behavior, impact, etc.). A simple presentation of measurements does not really contribute to our understanding on these events nor to the improvement of forecasting or mitigation activities. Similar measurements and observations are routinely performed worldwide. For example the origin and the evolution of this specific event has been forecasted by operational atmospheric dust models (e.g. http://www.bsc.es/ess/bsc-dust-daily-forecast) so there is really no need to perform HYSPLIT back trajectories.

Response: In this revision we have clearly stated our research objective in the beginning of the last introduction paragraph, which is to "picture a comprehensive view of dust event using different satellite and ground measurements with a recent heavy dust storm over northern China and southern Mongolia from 3 to 8 May 2017 as an example". Note the reviewer 2 commented that "...the authors combine advantages of satellite data and ground-based data, giving readers a comprehensive and detailed view for this dust event, including its transport trajectory, horizontal and vertical properties of storm,

and its influence on aerosol properties. It can be expected that the study provides a useful contribution to dust transport and related to this Journal." And the reviewer #3 stated that "the authors used diverse sources of observations to generate the knowledge on origin, timing and spatial coverage of the dust storm, overcoming setbacks of one observational system with other sources of measurements, leaving no room for uncertainties in created hypothesis on this event."

We have also changed the title and abstract to reflect clearly the objective of this study. We made full use of diverse sources of observations to capture the spatial-temporal distribution of the dust storm, as a single observational system is usually unable to provide such information. Observations from both polar-orbit and geostationary satellites, from active and passive remote sensing, and from ground based measurements were used. In addition, intensive ground-based PM measurements are not derived from the optical method and thus free from the influences of clouds and can even provide measurements during night-time. This complements to the blind areas of satellite observation affected by cloud and in the night time.

We agreed with the reviewer that the operational atmospheric dust models can provide dust-forecast. For example, there are four forecast models from MACC-ECWMF, NGAC-NCEP, KMA (Korea Meteorological Administration), and CMA (China Meteorological Administration), respectively, for the dust storm forecasting for East Asia. However, as stated above, the purpose of this study to demonstrate that combining different models/observations can capture a comprehensive view of dust event. In addition, this case study presented here may be used "in further numerical models development and verification" as stated by Reviewer #3.

Dear Editor and reviewers,

Thanks for the valuable comments, which help to improve significantly the quality of the paper. In this revision, we addressed the majority of the reviewer comments especially in terms of the study objective, figure clarity and sentence grammars rephrased. The detailed replies are listed below point by point in red.

Best regards,

Lu She on behalf of all authors

**Interactive comment on "Investigation of severe dust storms over the**

**Pan-Eurasian area using multi-satellite observations and ground-**

**based measurements" by Lu She et al.**

**Anonymous Referee #2**

In this paper, authors use multi-satellite observations and ground-based measurements to analyses a strong dust storm occurred in East Asia during 3 - 8 May 2017, the longdistant transport of the strong dust storms and the properties of dust aerosols were analyzed. The paper investigated the sources and different transport directions of the dust storms from different satellite observation (OMI, CALIPSO, and AHI) and particle matter (PM) measurements from ground-stations, and the HYSPLIT model were used to calculate the backward trajectories of air masses. The aerosol properties and its variation during no-dusty and dusty days were compared using AERONET data. The paper is clearly structured and logical. The authors combine advantages of satellite data and ground-based data, giving readers a comprehensive and detailed view for this dust event, including its transport trajectory, horizontal and vertical properties of storm, and its influence on aerosol properties. It can be expected that the study provides a useful contribution to dust transport and related to this Journal. However, the language of the paper requires some improvements. There are some sentences that are unclear or too long to follow. There are also some redundancies that should be removed. But I realize the authors' first language is not English, and this is not a criticism of them. I would recommend publication if my following comments/suggestions can be adequately addressed. Some comments and questions are given as follows:

Major comments:

1. The core of this paper, in my opinion, is to clearly describe the dust transport process

and the dust affected areas. The authors used long length to explain the transport of dust storm based on multi observations, but it would be better to see a more compact analysis with clearer connections between different observations.

Response: This has been improved in the revision in two aspects: (1) The depiction of dust transports revealed from different satellite time series observations was shortened as most of them exhibit the same pattern. (2) We added several sentences to illustrate the correspondence among different satellite observations. For example, the OMI observations and the CALIPSO were used together to confirm the dust area. The backward trajectories from the HYSPLIT were used to determine the dust source and the dust storm propagation direction. The PM measurements were collected as an effective complement for cloud affected area in the satellite observations, e.g., the south dust transport direction was mostly affected by cloud.

2. The authors should define the scientific aims of this study in more detail than what is done in the last paragraph of the introduction.

Response: The objective of this paper has been stated in the beginning of the last introduction paragraph as "This study tried to picture a comprehensive view of dust event using different satellite and ground measurements with a recent heavy dust storm over northern China and southern Mongolia from 3 to 8 May 2017 as an example."

The objective is based on the observation that "...few studies have been carried out to fully examine the source, distribution, transport, optical properties of the dust storm. This is possibly because each observation system can only characterize one or several aspects of them."

3. The authors point out that the dust transported to Korean Peninsula and Japan, but I don't see much analysis supporting these findings, especially for Korean Peninsula. Please check this claim more carefully.

Response: Thanks for this reminder. We have added the following sentence in the first paragraph of Section 3.1. "Furthermore, there is a small portion of the high AI values in the Japan Sea on 7 May (Fig. 2e) indicating that there is a second dust transport path of all the way east and the Korean Peninsula and Japan were affected."

We have revised the following sentence in the third paragraph of the Section 3.1 as below:

"The HYSPLIT backward trajectory analysis revealed that the air masses that reached the Bering Sea (Fig. 5a), the Kamchatka Peninsula (Fig. 5b), the Sea of Okhotsk (Fig. 5c), and the Japan Sea (Fig. 5d), originated from the Gobi Desert."

4. The authors have also analyzed the aerosol property variation using four AERONET

sites measurements. The variations in the AOD (440 nm) and Ångström exponent at four sites are shown, but why just show the VSD and SSA at Beijing and Baotou, what about Xuzhou-CUMT and Ussuriysk?

Response: There was no VSD and SSA inversion product for Xuzhou-CUMT and Ussuriysk sites during May 3 - 8, 2017. We have specified this in Fig. 14 caption.

5. There are some sentences and points which are confusing and invalid, even misleading readers. I suggest authors polish those important sentences to make your analysis more useful and clear.

Response: We have throughout checked the paper and revised our English writing carefully.

6. It is hard to read the figures, because some figures are heavily digitized. So I suggest authors to re-plot them or upload un-compressed manuscript.

Response: This has been improved. Details are in blow responses.

Detailed comments:

1. Line 53, 'mm' should be 'µm'

Response: This has been corrected.

2. Fig.2, suggest to use "brown" or other color scheme to represent the UV\_AI within

0-1. In addition, the labels on the color bar almost cannot be read! Please enlarge.

Response: This has been corrected. The labels have been enlarged and the color scheme has been modified so that the extreme high AI values pop up better.

3. Fig.3 the PM sites cannot be read. Please enlarge. We can barely read what is written.

Response: The letters have been enlarged. Note the contents of this Figures have been moved to other figures and the PM sites the reviewer concerned were now shown in Fig. 9 with enlarged labels.

4. Fig.4 the orbit tracks is not clearly depicted, please enlarge or just deleted, as the trajectories have been shown in Fig.3

Response: The orbit tracks have been moved to Fig. 2 and were shown with a clear dark blue color.

5. Line 191, 'over the region of northern China on 6 May', it seems that the overpass trajectory of 6 May didn't pass over northern China, see fig.4d. Please check it

Response: It should be 5 May, and we have corrected it in this revised version.

6. Line 212-215, sentence structure needs to be revised.

Response: We have changed this sentence to be "The HYSPLIT backward trajectory analysis revealed that the air masses that reached the Bering Sea (Fig. 5a), the Kamchatka Peninsula (Fig. 5b), and the Sea of Okhotsk (Fig. 5c), and the Japan Sea (Fig. 5d), originated from the Gobi Desert.".

7. Line 231, 'true-colour' should be 'true color'

Response: This has been corrected.

8. Fig.6 and Fig.7 are somewhat blurred, it's hard to tell the 'dust clusters' that described in line 235, as well as the dust transport.

Response: This has been improved. We have marked out the 'dust clusters' in Fig. 6 and Fig.7, and the dust transport direction have been marked with arrows.

9. Line 233-247: This section is a bit confusing, it should be rephrased to make it clearer.

Response: This part has been rephrased in the revised version.

10. Line 262 'caused a high PM10 concentration (>500) in south-central China (e.g., Hunan Province)' It would be better to specify the fig.- rather 'Hunan province', as it's not shown on the map, it is just a new city name to reader.

Response: This has been improved.

11. The authors should clearly conclude the transport process of dust, including different transport directions in 'Result' section.

Response: This has been improved. The dust storm propagation in different directions has been added in Fig.1. Furthermore, different data sources have different advantages to reveal the propagation in different directions. Consequently, we have following sentences in the results section

"The OMI-AI revealed one of the long-distance transport path of the strong absorbing aerosols that originated from the Gobi Desert and moved towards the east and then northeast (hereafter referred to as northeast direction for simplicity)."

"Furthermore, there is a small portion of the high AI values in the Japan Sea on 7 May (Fig. 2e) indicating that there is a second dust transport path of all the way east and the Korean Peninsula and Japan were affected."

"Part of the dust plume over southwestern Inner Mongolia moved along the edge of the Qinghai-Tibet Plateau and then finally reached the northern Sichuan basin (Fig. 6c), revealing the third path of the dust transport. This path of the dust transport is not revealed in the OMI AI time series maps possibly because the dust in this path is not very severe. ... High-frequency observations from the AHI presented more information about this severe dust storm, revealing multi-plumes propagation and several different transport directions, including southeastward, eastward and northeastward. The longest-distance transport occurred in the northeastward direction, as OMI-AI and

CALIPSO-VFM illustrated in the previous section, and finally arrived at the Bering Sea."

"In this section, the temporal variations in the PM2.5 and PM10 mass concentrations over mainland China were analysed and the third path of the dust transport, i.e., towards south, is obvious."

Dear Editor and reviewers,

Thanks for the valuable comments, which help to improve significantly the quality of the paper. In this revision, we addressed the majority of the reviewer comments especially in terms of the study objective, figure clarity and sentence grammars rephrased by a colleague living in an English-speaking country. The detailed replies are listed below point by point in red.

Best regards,

Lu She on behalf of all authors

**Interactive comment on "Investigation of severe dust storms over the**

**Pan-Eurasian area using multi-satellite observations and ground-**

**based measurements" by Lu She et al.**

Anonymous Referee #3 Received and published: 5 July 2018

**General comments**

The Study presented in this manuscript analyze in details large-scale heavy dust storm during May 2017 over Asia. Airborne dust originated from Gobi desert dispersed in several dust plumes, which propagated for several days in different directions. The authors used diverse sources of observations to generate the knowledge on origin, timing and spatial coverage of the dust storm, overcoming setbacks of one observational system with other sources of measurements, leaving no room for uncertainties in created hypothesis on this event. Scientific significance, scientific and presentation quality are good. Presented subject is of great significance because of the popularity of the topic, large impact of dust on climate system, but still not well understood and poorly represented in numerical models. Case study described here may be well used in further numerical models development and verification, since it is hard to correctly capture and describe fully any dust storm. This reviewer recommends this manuscript for publishing, after consideration of the following comments.

**Specific comments**

1) The title mentions in plural "dust storms", but in the manuscript is analyzed one dust storm that dispersed in several dust plumes. In the text is also mixture in mentioning dust storm as single event and dust storms as plural. To avoid confusion the authors should decide to define this event as one dust storm that has divided in several dust plumes or to define this event as severe airborne dust transport, which consists of several dust storms with the same origin. This reviewer suggests defining described event as severe dust storm that has complex multi-plume propagation. Whatever the authors decide, title and the mentioning in the text of the manuscript should be changed accordingly. In the title should be the date of the event, to outstand that the study covers specific study case.

Response: The title has changed to be "Towards a comprehensive view of dust event from multiple satellite and ground measurements: exemplified by the East Asia May 2017 dust storm" in response to this and also to the Reviewer #1 and 2's concerns on the paper objective.

We have referred to the event as one dust storm that has divided in several dust plumes.

2) In the manuscript there is no analysis of meteorological parameters to be able to understand the atmospheric conditions that produced this severe large-scale dust storm.

It is very important, to fully understand the event, to provide information about synoptic situation. To simplify this request it is enough to add the information on surface wind velocity and direction in the source region at the time of dust emission (or surface wind field), and to provide wind fields at representative height and/of geopotential heights (for example 500mb level) in representative times for later days. This would additionally explain the atmospheric circulation that carried dust particles. Data can be used from reanalysis fields.

Response: We have added the spatial distribution of wind velocity and direction, and geopotential height fields (Fig.3). Related analysis has also been added.

3) Add information about source of input data for HYSPLIT model that produced backward trajectories.

Response: Information about the input data for HYSPLIT mode have been added the revised version in section 2.7.

4) It would be very useful to add an image that presents hypothesis about dust storm propagation in different directions (or mark with arrows in Fig. 1), which is proved using many observations. It is hard to follow in case the geography of the region is not well known.

Response: This has been improved. See Fig.1.

Technical corrections

1) line 26: change "10 mm" in "10 \_m"

Response: This typo has been corrected.

2) line 55: change "10 mm" in "10 \_m"

Response: This typo has been corrected.

3) line 59: change "Many studies have been carried out to study different aspects of dust plumes from deserts using..." in "Many studies have been carried out to study different aspects of airborne dust transport from deserts using..."

Response: This sentence has been rephrased as "Many literatures have studied desert dust from different perspectives using different satellite data, ground-based observations and model simulations (Badarinath et al., 2010; Wang et al., 2013; Teixeira et al., 2016)".

4) line 120: change "that can used to..." in "that can be used to..."

Response: This has been corrected.

5) line 123: change "It has been suggested that..." in "It has been evaluated that..."

Response: This has been corrected.

6) line 136/137: change "The inversion products includes both microphysical parameters ..." in "The inversion products include both, microphysical parameters ..."

Response: This has been corrected.

7) line 146: change in "... during both, day-time and night-time."

Response: This has been corrected.

8) line 149/150: change in "...were collected to evaluate the dust-affected areas and to further analyse the transport of the dust plume."

Response: This has been corrected.

9) line 172: is it correct to have easterly wind in "... swept through the North China plain on 4 May 2017 due to a strong easterly wind,..."? It is not likely to have east wind, maybe west wind, which means that circulation was eastward?

Response: This has been corrected. It should be west wind, and we have also added the information about wind direction, see Fig. 3

10) line 215: exclude "The" at the beginning of "The This result is ..."

Response: This has been corrected.

11) line 254: In the following sentence "Fig. 8 depicts the PM10 concentration distribution..." add an information about PM data values, are they hourly average of what? how many stations are considered?

Response: More information about PM data has been added in the revised version. The PM values are hourly average, and all the valid observations from 1350 stations located in mainland China were used. The PM values are real-time measurement, the air stations make the measurements every hour. 12) Letters in some Figures are too small, and require large zoom to be readable, especially Fig. 2, Fig. 4 and Fig. 9. If possible, use different rearrangement of plots and landscape mode.

Response: This has been corrected. The letters in figures were enlarged, and the figures have been rearranged in the revised version.

- 1 Investigation Towards a comprehensive view of severe dust
- 2 storms over the Pan-Eurasian area using multi-event from
- 3 multiple satellite <del>observations</del> and ground-based
- 4 measurements: exemplified by the East Asia May 2017 dust
- 5 storm
- Lu She1,3,4,5, Yong Xue2,3, Jie Guang3, Yaihui Che3, Cheng Fan3, Ying Li3, Yanqing
  Xie3
- 1College of Resources and Environmental Science, Ningxia University, Ningxia 750021, Ningxia
  9 Province, China
- 2Department of Electronics, Computing and Mathematics, College of Engineering and Technology,
- 11 University of Derby, Derby DE22 1GB, UK
- 12 3Key Laboratory of Digital Earth Science, Institute of Remote Sensing and Digital Earth, Chinese
- 13Academy of Sciences, Beijing 100094, China
- 4Ningxia Key Laboratory of Resources Assessment and Environmental Regulation in Arid Regions,
- 15 Yinchuan 750021, Ningxia Province, China
- 16 5China-Arab Joint International Research Laboratory for Featured Resources and Environmental
- 17 Governance in Arid Regions, Yinchuan 750021, Ningxia Province, China.
- 18
- 19 *Correspondence to*: Professor Yong Xue (y.xue@derby.ac.uk)

20 Abstract. The deserts in One or several aspects of the source, distribution, transport, optical properties 21 of airborne dust have been characterized using different types of satellite and ground measurements each 22 with unique advantages. In this study, a dust event occurred over the East Asia are one of the most 23 influential mineral dust source regions in the world. Large amounts of dust particles are emitted and 24 transportedarea in May 2017 was exemplified to distant regions. A superdemonstrate how all the above mentioned aspects of a dust storm characterized by long distance transport occurred over the Pan-25 26 Eurasian Experiment (PEEX) area in early May 2017. In this study, multi-satellite/sensor observations 27 and ground event can be pictured by combining the advantages of different satellite and ground 28 measurements. The used data included the Himawari-8 satellite Advanced Himawari Imager (AHI) true-29 colour images, the Cloud-Aerosol Lidar and Infrared Pathfinder Satellite Observation (CALIPSO) 30 Cloud-Aerosol Lidar with Orthogonal Polarization (CALIOP) aerosol vertical profiles, the Aura satellite 31 Ozone Monitoring Instrument (OMI) aerosol index images, and the ground based measurements 32 combined with the HYbrid Single Particle Lagrangian Integrated Trajectory (HYSPLIT) model were

33 used to analyse the dynamical processes of the origin and transport of the strong dust storm. The optical 34 and microphysical properties of the dust particles were analysed using Aerosol Robotic Network 35 (AERONET) aerosol properties and the ground station particulate matter (PM) measurements. From the 36 multi-satellite/sensor (AHI, CALIOP and OMI) time series observations, the dust storms were 37 suggestedstorm was found to have originated originate from the Gobi Desert on the morning of 3 May 38 2017, and it transported dusttransport northeastward to the Bering Sea, eastward to the Korean Peninsula 39 and Japan, and southward to southern Centralsouth-central China. The air quality in China drastically 40 deteriorated as a result of this heavy dust storm; drastically: the  $PM_{10}$  (particulate matter less than PM < 1041 mmµm in aerodynamic diameter) concentrations measured at some air quality stations located in northern 42 China reached 4000-4333 µg/m3. During the dust eventAt the AOE Baotou, Beijing, Xuzhou-CUMT, 43 and Ussurivsk AERONET sites, the maximum AODaerosol optical depth values reached 2.96, 2.13, 3, 44 2.3, 2.87, and 0.65 with sharp drops in and the extinction Angström exponent (EAE) dropped to 0.023, 45 0.068, 0.0317 and 0.097 at AOE\_Baotou, Beijing, Xuzhou CUMT, and Ussuriysk, respectively. The dust 46 storm introduced great variations in thealso induced unusual \_-aerosol property, causing totally different 47 spectral single-scattering albedo (SSA) and volume size distribution (VSD). The combined observations 48 revealed comprehensive information about the dynamic transport of dust and the dust affected regions, 49 and the effect of dust storms on the aerosol properties.

50

**51 1. Introduction**

[revised manuscript text omitted]